# Extracellular Vesicle miR-200c Enhances Gefitinib Sensitivity in Heterogeneous EGFR-Mutant NSCLC

**DOI:** 10.3390/biomedicines9030243

**Published:** 2021-02-28

**Authors:** Chien-Chung Lin, Chin-You Wu, Joseph Ta-Chien Tseng, Chun-Hua Hung, Shang-Yin Wu, Yu-Ting Huang, Wei-Yuan Chang, Po-Lan Su, Wu-Chou Su

**Affiliations:** 1Department of Internal Medicine, National Cheng Kung University Hospital, College of Medicine, National Cheng Kung University, Tainan 704, Taiwan; joshcclin@gmail.com (C.-C.L.); wusy.tw@gmail.com (S.-Y.W.); n010230@gmail.com (Y.-T.H.); polan.750317@gmail.com (P.-L.S.); 2Institute of Clinical Medicine, College of Medicine, National Cheng Kung University, Tainan 704, Taiwan; lineage771029@gmail.com; 3Department of Biochemistry and Molecular Biology, College of Medicine, National Cheng Kung University, Tainan 701, Taiwan; 4Department of Biotechnology and Bioindustry Sciences, National Cheng Kung University, Tainan 701, Taiwan; josephtseng777@gmail.com; 5Center of Applied Nanomedicine, National Cheng Kung University, Tainan 704, Taiwan; bobofranzi@gmail.com; 6Department of Internal Medicine, An Nan Hospital, China Medical University, Tainan 709, Taiwan; lysergsite@gmail.com

**Keywords:** extracellular vesicles, miRNA, EGFR mutation, heterogenous, NSCLC, EGFR-TKI

## Abstract

Intratumoral heterogeneity in epidermal growth factor receptor (EGFR)-mutant mutant non-small-cell lung cancer (NSCLC) explains the mixed responses to EGFR-tyrosine kinase inhibitors (TKIs). However, some studies showed tumors with low abundances of EGFR mutation still respond to EGFR-TKI, and the mechanism remained undetermined. Extracellular vesicles (EVs) can transmit antiapoptotic signals between drug-resistant and drug-sensitive cells. Herein, we profiled EVs from EGFR-mutant cells to identify a novel mechanism explaining why heterogenous EGFR-mutant NSCLC patients still respond to EGFR-TKIs. We first demonstrated that the EVs from EGFR-mutant changes the wild-type cells’ sensitivity to gefitinib by adding EV directly or coculturing EGFR wild-type (CL1-5) cells and EGFR-mutant (PC9) cells. In animal studies, only the combined treatment of PC9 EV and gefitinib delayed the tumor growth of CL1-5 cells. MicroRNA analysis comparing EV miRNAs from PC9 cells to those from CL1-5 cells showed that mir200 family members are most abundant in PC9 EVs. Furthermore, mir200a and mir200c were found upregulated in plasma EVs from good responders to EGFR-TKIs. Finally, the transfection of CL1-5 cells with miR200c inactivates downstream signaling pathways of EGFR, the EMT pathway, and enhances gefitinib sensitivity. Overall, our results suggest that in heterogeneous EGFR-mutant NSCLC, tumor cells transmit EV miRNAs that may affect sensitivity to EGFR-TKIs and provide potential prognostic biomarkers for EGFR-mutant NSCLC.

## 1. Introduction

Intratumoral heterogeneity in epidermal growth factor receptor (EGFR)-mutant non-small-cell lung cancer (NSCLC), which has been found to range from 13.9 to 27%, explains the mixed-response phenomenon and results in acquired resistance to tyrosine kinase inhibitors (EGFR-TKIs) [1]. However, some studies have also demonstrated that heterogeneous EGFR-mutant lung cancer patients can still exhibit good responses to EGFR-TKIs. For example, de Biase et al. [2] used a highly sensitive next-generation sequencing (NGS) platform to detect the percentage of mutated EGFR alleles and found a correlation (*p* = 0.068) between the percentage of mutated alleles and the responsiveness to TKIs. However, no correlation was noted between the percentage of mutated alleles and progression-free survival (PFS, *p* = 0.268) or overall survival (OS, *p* = 0.708), and even patients with low percentages (less than 50%) still exhibited partial response or stable disease following EGFR-TKI treatment [2]. In another larger cohort study using direct DNA sequencing and an amplification refractory mutation system (ARMS) analysis to determine the percentage of mutation-positive tumors, EGFR mutations were detected in 51 samples (51%) by both sequencing and ARMS analysis (high-abundance group), while 18 of the other 49 samples that were EGFR-mutation negative according to the sequencing were positive according to the ARMS analysis (low-abundance group). Though the patients with high abundances of EGFR mutations were found to have a better mean PFS duration than those with low abundances (11.3 versus 6.9 months, *p* = 0.014), there were no significant differences between the high-abundance group and low-abundance group patients in terms of objective response rate (ORR 62.7% versus 44.4%, *p* = 0.1766) or OS (15.9 versus 10.9 months, *p* = 0.062) [3]. However, the mechanisms explaining why EGFR-TKIs can be effective in cases of heterogeneous NSCLC with low abundance of EGFR mutations remain unclarified.

Cells release different types of extracellular vesicles (EVs), including microvesicles, which bud from the cellular plasma membrane, and exosomes, which are derived from multivesicular bodies [4]. EVs have a key role in regulating cell–cell communication through the transfer of molecular cargo, including proteins and miRNA [5]. The landmark study by Valadi et al. [6] revealed that EV mRNA from mast cells can be transported to recipient cells and then translated into proteins with biological functions. Zomer et al. [7] further demonstrated that T47D mammary tumor cells with low malignancy can take up EVs derived from the more malignant MDA-MB-231 cells and then display increased migratory ability. A recent review article highlighted the findings that these EVs can transfer drug resistance by mechanisms that include antiapoptotic signaling and increased DNA repair capability or deliver ABC transporters from drug-resistant cells to drug-sensitive cells [8]. Some studies have also demonstrated that EVs from EGFR-mutant lung cancer cells may affect sensitivity to EGFR-TKIs or chemotherapy [9,10]. However, whether EVs from EGFR-mutant cells can mediate EGFR-TKI sensitivity in heterogeneous, treatment-naïve NSCLC with a low percentage of EGFR mutations remains unclear. Among the cargo of EVs, miRNAs are small noncoding RNAs that control gene expression post-transcriptionally, and EVs can increase the therapy resistance of the donor cell by delivering miRNAs [8]. Furthermore, there is increasing evidence suggesting that miRNAs can serve as valuable pathological and therapeutic biomarkers in EVs because microRNAs can alter global protein synthesis, be released from cancer cells into the circulation, and accumulate in EVs protected from cleavage by RNases [8]. Specifically, the identification of EV miRNAs associated with EGFR-TKI sensitivity can help predict EGFR-TKI responses in patients receiving EGFR-TKI treatment. Under this scenario, we hypothesized that the transfer of EV miRNAs between EGFR-mutant and wild-type cancer cells mediates the response of EGFR-TKI in heterogeneous EGFR-mutant NSCLC.

## 2. Materials and Methods

### 2.1. Cell Lines and Culture

The human lung cancer cell lines PC9 (EGFR exon 19 deletion) and CL1-5 were kindly provided by Dr. Chih-Hsin Yang and Dr. Pan-Chyr Yang (National Taiwan University College of Medicine, Taipei, Taiwan), respectively. A549 and H1299 cells were obtained from American Type Culture Collection (Rockville, MD, USA). A549 cells were maintained in F12K (Thermo Fisher Scientific, Waltham, MA, USA). PC9, H1299, and CL1-5 cells were maintained in Roswell Park Memorial Institute (RPMI) medium (Thermo Fisher Scientific, Waltham, MA, USA). All cell lines were verified by DNA STR and tested for mycoplasma infection.

### 2.2. The Ultrafiltration (UF) Method

The conditioned media and the clinical biofluids were centrifuged at 1000× *g* at 4 °C for 5 min, and then the supernatants were centrifuged again at 1000× *g* at 4 °C for 10 min. Next, the supernatants were passed through 0.22 μm filters, and the filtered supernatants were then subjected to EV isolation using 100 kDa Vivaspin nanomembrane concentrators (Sartorius Stedim Biotech, Germany). After being centrifuged at 3000× *g* at 4 °C for 10–90 min, the EV and the non-EV samples were collected from the retention and flow-through portions, respectively. The EV samples were washed once with 0.22 μm-filtered 1X PBS. For Western blot analysis, the non-EV samples were then concentrated using 10 kDa Vivaspin nanomembrane concentrators and were washed once with 0.22 μm-filtered 1X PBS [11].

### 2.3. Transmission Electron Microscopy (TEM)

For TEM, the EVs were mixed with 4% paraformaldehyde/1% glutaraldehyde (Merck & Co., Inc., Darmstadt, Germany). The samples were washed with MQ water using Vivaspin nanomembrane concentrators (Sartorius Stedim Biotech GmbH, Göttingen, Germany). The samples were loaded onto carbon-coated formvar grids (Ted Pella Inc., Redding, CA, USA) and negatively stained with 0.2% uranyl acetate (Merck & Co., Inc., Darmstadt, Germany) for 3 min. The samples were then air-dried and examined with a JEM-1400 transmission electron microscope or JEM-2100F CS STEM electron microscope (JEM- 2100F; JEOL, Tokyo, Japan).

### 2.4. Size Distribution Measured by Nanoparticle Tracking Analysis

For TEM, the EVs were mixed with 4% paraformaldehyde/1% glutaraldehyde (Merck & Co., Inc., Darmstadt, Germany). The samples were washed with MQ water using Vivaspin nanomembrane concentrators (Sartorius Stedim Biotech, Germany). The samples were loaded onto carbon-coated formvar grids (Ted Pella Inc., Redding, CA, USA) and negatively stained with 0.2% uranyl acetate (Merck & Co., Inc., Darmstadt, Germany) for 3 min. The samples were then air-dried and examined with a JEM-1400 transmission electron microscope or JEM-2100F CS STEM electron microscope (JEOL).

### 2.5. Cell/EV Lysis and Western Blot Analysis

For cell/EV lysis, the harvested cells/EVs were incubated on ice in a whole-cell-extract lysis buffer and centrifuged, and the protein concentration was then measured by a Bradford assay (Biorad Laboratories, CA, USA). For the Western blot analysis, the lysates were then boiled with sample buffer before being separated on SDS-polyacrylamide gels. Proteins were transferred to polyvinylidene difluoride membranes (Millipore Corp., Billerica, MA, USA) and blocked with 5% nonfat milk/TBST buffer. Using an electrochemiluminescence kit (Amersham Pharmacia Biotech, Piscataway, NJ, USA), we detected the binding of specific antibodies. The EV marker-specific antibody CD63 was purchased from Santa Cruz Biotechnology, while the ER marker GRP78-specific antibody was purchased from Abcam (Cambridge, UK). To investigate changes in the activity of downstream EGFR signaling after treatment, the following antibodies were used: (1) anti-phospho-Stat3 (Tyr705) (Cell Signaling, Danvers, MA, USA), (2) anti-Stat3 (BD Biosciences, San Jose, CA, USA), (3) anti-actin (Millipore), (4) anti-phospho-Akt (Ser473) (Cell Signaling), (5) anti-Akt (Cell Signaling), (6) anti-phospho-Erk (R&D, Minneapolis, MN, USA), (7) anti-Erk (Santa Cruz Biotechnology, Santa Cruz, CA, USA), (8) anti-E-cadherin (Cell Signaling), and (9) anti-Zeb1 (Santa Cruz Biotechnology, Santa Cruz, CA, USA).

### 2.6. EV Staining, Immunofluorescent Images, and Live Imaging

EVs from PC9 cells were labeled using PKH26 (red) membrane-binding fluorescent labels according to the manufacturer’s recommendations (Sigma-Aldrich, Allentown, PA, USA). CL1-5 cells seeded on chamber slides (Thermo Scientific Inc., Rochester, NY, USA) were incubated at 37 °C with labeled EVs at a concentration of 1 µg EV/10,000 cells or as described. Uptake was stopped by washing and fixation in 4% paraformaldehyde for 10 min. Labeled PC9 cells were generated after incubation of a standard culture-cell monolayer for 12 h with PKH26-labeled EVs. The slides were then blocked in PBS-T containing 5% normal donkey serum for 1 h. The cells were incubated with antitubulin antibody (Santa Cruz Biotechnology) diluted in PBS 1% BSA for 1 h at 37 °C. Coverslips were washed three times with PBS and treated with Alexa Fluor 488 goat antimouse IgG (Invitrogen) for 30 min at 37 °C. The nuclei were visualized by 4,6-diamidino-2-phenylindole staining (DAPI, Sigma-Aldrich, Saint Louis, MO, USA). The fluorophores were excited by lasers at 405, 488, and 543 nm and detected by a scanning confocal microscope (FV-1000, Olympus Corp, Tokyo, Japan). The resulting three-color images were exported as TIFF images. For live imaging of EV uptake, coverslips containing CL1-5 cells were placed in a temperature-controlled live-cell imaging chamber (the cube/the box, Live Imaging Services) and observed under imaging medium containing PKH26-labeled EVs from PC9 cells using a Leica confocal microscope DMIRE2.

### 2.7. DNA Quantification and Droplet Digital PCR Analysis

To determine the transportability of the DNA in EVs between EGFR-mutant and EGFR wild-type cell, EGFR wild-type cells were incubated with 200 μg/mL EVs derived from PC9 cells for 72 h before DNA extraction. DNA of each cell line and PC9 EVs were extracted using a QIAamp DNA mini kit (QIAGEN, Hilden, Germany). Total DNA from PC9 EVs, PC9 cells, EGFR wild-type cells (A549, CL1-0, CL1-5 and H1299), and EGFR wild-type cells incubated with PC9 EVs were quantified using a fluorescence absorbance PicoGreen assay (Invitrogen). Quant-iT™ PicoGreen^®^ dsDNA reagents and kits (Invitrogen) were used according to the manufacturer’s protocol. Samples were excited at 480 nm, and the fluorescence emission intensity was measured at 520 nm using an EnSpire^®^ multimode plate reader (PerkinElmer, Villebon-sur-Yvette, France). TaqMan PCR mixtures were assembled from a 2× ddPCR MasterMix (Bio-Rad Laboratories, Inc., Hercules, CA, USA) and custom 40× TaqMan probes/primers made specifically for each assay. Analysis of the ddPCR data was performed with the QuantaSoft analysis software (Bio-Rad Laboratories, Inc., Hercules, CA, USA) that accompanied the droplet reader, as was carried out in a previous study [12].

### 2.8. Combination Treatment with Gefitinib and EVs and MTT Assays

The EGFR wild-type cells were seeded onto 96-well plates at a density of 2500 cells per well and incubated with indicated concentration of EVs (EVs from themselves or PC9) for 48 h. The cells were then treated with the indicated concentration of gefitinib with EV for an additional 72 h. Then 20 μL of MTT solution (5 mg/mL in phosphate-buffered saline (PBS)) was added to each well and then continued for a further 4 h at 37 °C. Cell viability was calculated as a ratio between treated (sample) and untreated (control).

### 2.9. The Coculture System, GW4869 Treatment, and MTT Assays

The coculture system was performed using Transwell™ supports (pore size: 1 μm) inserted into the wells of 24-well plates. PC9 cells or EGFR wild-type cells (4 × 10^5^ cells/well) were placed in the bottom chamber and EGFR wild-type cells (3 × 10^3^ cells/well) were placed in the upper chamber. When PC9 cells were placed in the bottom chamber, DMSO or GW4869 (2.5 μM) was added the bottom chamber in coculture with EGFR wild-type cells. After 72 h of coculture, EGFR wild-type cells in the upper chamber were treated with gefitinib for 72 h, and cell viability was examined by MTT assay.

### 2.10. MicroRNA Transfection into Cells and EV and MTT Assays

Lung cancer cells were transfected with 100 nmol/L pre-miR-200c (hsa-miR-200c-3p, #MC11714) or control miR (miR-scrambled) using Lipofectamine RNAiMAX (Thermo Fisher Scientific Inc., Waltham, MA, USA), as was carried out in a previous study [13]. The transfected cells were used for the MTT assay (Sigma, St. Louis, MO, USA) or Western blotting 48 h after the final transfection.

### 2.11. Animal Model

To evaluate the effect of administration of PC9 EVs on EGFR-TKI sensitivity of CL1-5 in vivo, a calculated final total of 1 × 10^6^ CL1-5 cells in 100 µL RPMI were injected subcutaneously into 7-week-old BALB/cAnN.Cg-Foxnlnu/CrlNarl female mice (supplied and approved at Feb 2018 by the Animal Center at the College of Medicine, National Chen-Kung University, Tainan, Taiwan, IACUC-107021). The calculation of the tumor volume was conducted according to the following formula: volume = length × width^2^ × 0.5. When the tumor sizes were approximately 100 mm^3^, xenograft mice were randomized into four groups: namely, the control, EV, gefitinib, and combination groups. According to animal welfare, ethics, and the 3Rs, mice were divided into four groups with three mice per group. Gefitinib (50 mg/kg/day) was prepared in drinking water and fed through oral administration. For the EV and combination groups, tumor-bearing mice were intratumorally injected with PC9 EVs at a dose of 20 μg EV proteins twice per week. Tumors were measured twice per week by a technician blinded to the experimental setup.

### 2.12. Patient and Sample Processing

Peripheral blood samples were taken from patients with late-stage EGFR-mutant NSCLC before receiving EGFR-TKI treatment. Peripheral blood samples from these specimens were collected in 10 mL plasma separator tubes. Within 2 h after collection, the plasma samples were fractioned into multiple aliquots after centrifugation and then stored at −80 °C until use. Informed consent was obtained from all study participants prior to blood draw. Each patient provided written informed consent. The protocol for this study (IRB-100-034) was approved by the Institutional Review Board of National Cheng Kung University Hospital.

### 2.13. EGFR-Mutation Abundance Evaluation Using Real-Time PCR and Immunohistochemistry Staining

The presence of EGFR mutations was determined using the EGFR polymerase chain reaction (PCR) kit (EGFR RUO Kit) and therascreen^®^ EGFR RGQ PCR Kit (EGFR IVD Kit, Qiagen, Manchester, UK) [14]. The difference in the cycle threshold (Ct) value of the mutant-versus-internal control (ΔCt) was used to determine the mutation status and relative abundance of EGFR mutation [15]. Immunohistochemical stains were performed on de-paraffinized, rehydrated tumor and using rabbit monoclonal antibodies that are specifically against EGFR E746-A750del (Cell Signaling Technologies (CST), Danvers, MA, USA) at 1:100 dilutions.

### 2.14. EV RNA Isolation from Blood and Micro-RNA Profiling

For EV isolation from plasma, plasma was first defibrinated with thrombin at room temperature for 10 min and then centrifuged to collect the supernatant. The serum-like supernatant was treated with ExoQuick exosome precipitation solution (SBI, Mountain View, CA, USA) to precipitate exosomes for 30–60 min at 5 °C. The vesicle pellets were dissolved in PBS, and exosome RNA was extracted immediately by TRIsureTM (Bioline, London, UK). Small RNAs were purified by the Direct-zol™ RNA MicroPrep Kit (Zymo Research, Irvine, CA, USA). For micro-RNA profiling, micro-RNA library preparation was performed with the NEBNext^®^ Small RNA Library Prep Kit (New England Biolabs, Hitchin, UK). In brief, adapters were ligated to the small RNAs, and cDNA synthesis was performed by reverse transcription. Following PCR amplification of the adapter-ligated cDNA, the libraries were size-selected on a polyacrylamide gel and purified with the AMPure XP system (Beckman Coulter, CA, USA). The quantified libraries were sequenced with the Illumina sequencing platform following the manufacturer’s instructions.

### 2.15. Statistical Analysis

The data collected were analyzed with Prism 6 (GraphPad Software for Science, Inc., San Diego, CA, USA) to determine statistically significant differences (*p* < 0.05). Student’s *t*-test was used as indicated in the figure legends.

## 3. Results

### 3.1. Characterization of Extracellular Vesicles Released from EGFR-Mutant Cells and Their Transfer to EGFR Wild-Type Cells

We first verified the characteristics of EV using Western blot analysis, transmission electron microscopy (TEM), and NanoSight nanoparticle analyzer. EVs from PC9 (EGFR-mutant) cells were isolated from cell-free medium using ultrafiltration, as in a previous study [11]. As shown in Figure 1A, the exosomal marker protein CD63 was positively expressed in cell lysates and EVs, whereas the ER marker GRp78 (BiP) was negatively expressed in EVs. The morphology of the isolated EVs analyzed by TEM showed typical lipid bilayer membrane-encapsulated nanoparticles, and the sizes of these nanoparticles were approximately 100–200 nm. Further analysis by a NanoSight nanoparticle analyzer showed that the average size of the EVs was 190 nm, which was compatible with the size observed by TEM.

To examine whether the secreted EVs could be naturally taken up by recipient cells, we recorded the uptake of EVs. The isolated EVs from PC9 were subsequently stained with the red lipophilic dye PKH26 and introduced to CL1-5 cell cultures. After 24 h, we observed that the PC9 EVs were taken up by CL1-5 cells and were mainly localized in the perinuclear area of the cytoplasm, as shown by time-lapse laser scanning microscopy images (Figure 1B upper, Appendix A) and confocal microscopy (Figure 1B lower) respectively.

We next planned to confirm the EV cargo can be transferred to recipient cell and could be detected. Because a previous study showed that EVs carry EGFR-mutant DNA [16], we checked whether EGFR-mutant DNA could be transferred from EV to recipient cells. After introducing PC9 EVs to EGFR wild-type cell including A549, CL1-0, CL1-5, and H1299 for 72 h, we analyzed EGFR-mutant and wild-type DNA by digital PCR. In PC9 and PC9 EVs (lanes 9 and 10), we could detect EGFR-mutant DNA (blue spot, Exon 19Del DNA) and EGFR wild-type DNA (green spot); conversely, EGFR-mutant DNA could not be detected in EGFR-wild type cell (lanes 1, 3, 5, and 7). However, EGFR-mutant DNA could be detected in recipient EGFR-wild-type cell (lanes 2, 4, 6, and 8) after adding PC9 EVs (Figure 1C). The result implied that EVs may be a general mechanism to exchange molecular cargo between EGFR-mutant and EGFR wild-type cells.

### 3.2. The Uptake of PC9 EVs Affects the Sensitivity of Wild-Type EGFR to Gefitinib

The role of EVs in mediating EGFR-TKI sensitivity in heterogeneous EGFR NSCLC remains unknown. We hypothesized that EVs may mediate drug sensitivity in heterogeneous lung cancer via the transfer of cargo from EGFR-mutant cells to EGFR wild-type cells, thus contributing to the sensitivity to EGFR-TKIs observed in heterogeneous EGFR-mutant NSCLC. We first tested the sensitivities of PC9 and other EGFR wild-type cells to gefitinib. PC9 cells were sensitive to gefitinib in a dose-response manner using MTT assays (Figure 2A). Conversely, EGFR wild-type cells are resistant to gefitinib, and cytotoxicity effect was observed only in high dose of gefitinib (100 μM versus 0.1 μM in PC9 cells, Figure 2A).

We then checked if EVs from EGFR wild-type cells did not affect the sensitivity of EGFR wild-type cells to gefitinib. We collected EVs from EGFR wild-type cells and treated EGFR wild-type cells with 0.1 μM gefitinib and titrating dose of EV derived from these cells respectively. We verified EVs from EGFR wild-type cells did not affect the sensitivity of EGFR wild-type cells to gefitinib (Figure 2B).

Finally, we treated CL1-5 cells with PC9 EVs (200 μg/mL) and with dose titrations of gefitinib (from 0.01 μM to 0.1 μM). We found PC9 EVs can significantly enhance the cytotoxic effect of gefitinib under the combination of 200 μg/mL EV and 0.1 μM gefitinib (cell viability < 50%, Appendix A). We further verified the exposure of other EGFR wild-type cells (CL1-0 and H1299 cells) to PC9 EVs increased their sensitivity to gefitinib (Figure 2C). However, PC9 EVs did not affect the gefitinib sensitivity of A549 cells, which harbor Kras mutations, and tumors only rarely exhibit both Kras and EGFR mutations [17].

### 3.3. Coculture with PC9 Cells Sensitizes EGFR Wild-Type Cells to Gefitinib, and Inhibition of Exosome Secretion Reverses This Effect

To clarify whether PC9-derived EVs had the capacity to modulate sensitivity, coculture experiments were performed to properly mimic tumor heterogeneity. Indirect cocultures in which wild-type EGFR cells and mutant EGFR cells were grown separately but with a membrane through which interactions via EVs could still occur were performed. While indirect transwell coculture allows for the exposure of wild-type EGFR cells to the EVs secreted by mutant EGFR cells, it does not allow the two different cell lines to directly contact each other (Figure 3A). Again, EGFR wild-type cells showed resistance to gefitinib when they were plated into both compartments of the coculture plate and then treated with gefitinib at a concentration of 0.1 μM, which is higher than the IC50 value in PC9 cells (Figure 3A). We found that the coculture of CL1-5 or H1299 cells with PC9 cells led to increased sensitivity to gefitinib but not in that of A549 cells (Figure 3A). We next tested whether the blockade of exosome secretion could reverse the sensitivity to EGFR-TKIs in coculture with PC9 cells. GW4869 is an inhibitor of neutral sphingomyelinase, which converts sphingomyelin to ceramide and inhibits the secretion of EVs [18]. We first demonstrated, by MTT assay, that GW4869 inhibited the secretion of EVs without any effect on cell survival (Figure 3B). Coculture with PC9 cells sensitized CL1-5 cells to gefitinib, but pretreatment with GW4869 for 24 h reversed the sensitivity to EGFR-TKIs in the coculture with PC9 cells (Figure 3C).

### 3.4. EVs Derived from EGFR-Mutant Cells Inhibit EGFR Wild-Type Tumor Growth In Vivo

To further evaluate the role of EVs in gefitinib resistance, we harvested EVs from PC9 cells and evaluated their effect on the gefitinib sensitivity of EGFR wild-type cells in an orthopedic animal model. We collected CL1-5 cells and subcutaneously injected them into athymic nude mice. When the tumor sizes were approximately 100 mm^3^, we administered PC9 EVs (20 μg) or PBS twice per week intratumorally to the treatment and control groups, respectively. The mice in the treatment group were orally treated with gefitinib, as was carried out in a previous study [19]. As shown in Figure 4 (Appendix A), neither gefitinib nor EV treatment alone inhibited tumor growth compared to the control group. Only the combination treatment with EVs and gefitinib delayed tumor growth.

### 3.5. Micro-RNA Expression Profiles Are Significantly Different between EVs from EGFR-Mutant Cells and EVs from EGFR Wild-Type Cells

Increasing evidence has revealed that EVs mediate tumor heterogeneity by delivering miRNAs [20,21]; therefore, we focused our attention on the role of EV miRNAs in mediating intercellular communication and drug sensitivity in heterogeneous NSCLC. To elucidate the effect of EV intercommunication in heterogeneous EGFR-mutant NSCLC, we isolated EVs from PC9 cells and CL1-5 cells and performed a comparative analysis of their miRNA content (Figure 5A, Appendix A). Relative expression levels of the miRNAs are provided in Appendix A. Among the miRNAs detected, miR-200c was the most abundant miRNA identified in PC9 EVs, and miR-200a showed the highest fold change compared to that in CL1-5 EVs. The differential miRNA expression profiles were then analyzed by IPA software, and the results showed that the possible pathways mediated by the differentially expressed miRNAs were EGFR downstream signaling pathways, such as the PTEN-AKT, Stat3, and Erk pathways (Figure 5B,C).

### 3.6. The miRNA Profiles of Circulating EVs Are Significantly Different between EGFR-TKI Good Responders and Poor Responders and Similar to the miRNA Profiles Identified in PC9 EVs

The ability of EVs to actively travel from cancer cells to intercellular matrices to finally reach the circulation and their stable characteristics may allow EV miRNAs to serve as biomarkers. Based on the finding that miRNAs from PC9 EVs were associated with responses to EGFR-TKI treatment, we hypothesized that these miRNAs may also be detected in the human circulation system and associated with treatment responses. From April 2015 to August 2017, we prospectively collected plasma from patients with late-stage EGFR-mutant NSCLC before receiving EGFR-TKI treatment. Because previous phase III trials unequivocally demonstrated that EGFR TKIs provided a median PFS of 9.2–11.1 months [22], we decided to define patients with PFS longer than 12 months as good responders and those with less than 6 months as poor responders. PFS was calculated from the date of EGFR-TKI initiation until the date of radiological progression according to the response evaluation criteria in solid tumors (RECIST) v1.1 [23] or death. Among the 10 enrolled patients, five experienced disease progression within 6 months, and five developed acquired resistance after 12 months (Figure 6A, Appendix A). EV enrichment for miRNA profile analysis was conducted for these samples. The clinical characteristics of the patients are listed in Table 1. We identified 23 miRNAs with differential expression between good responders and poor responders (*p* < 0.05) (Figure 6B). Among the scanned miRNAs, four miRNAs were also found to be differentially expressed in the miRNA panel (a total of 136 miRNAs) by comparing EV miRNAs from PC9 cells to those from CL1-5 cells (Figure 6B). MiR-200a and miR-200c were found to be upregulated, and miR-210 and miR-758 were found to be downregulated in good responders compared to poor responders.

### 3.7. Transfection of miR-200c Inhibits Downstream Signaling Pathways of EGFR and Enhances Gefitinib Sensitivity of EGFR Wild-Type Cells

Based on the above comparison study, it was speculated that the identified miRNAs may mediate EGFR-TKI sensitivity in heterogeneous EGFR-mutant cells. As such, we investigated whether one of these miRNAs, miR-200c, might be involved in the pathways identified by IPA software in EGFR wild-type cells. Using Western blot analysis, we demonstrated that transfection of miR-200c and gefitinib treatment decreased the phosphorylation of Stat3 and Akt in CL1-5 cells (Figure 6C). In addition, members of the apoptosis pathway, such as caspase-3 and caspase-9, were also activated (Figure 6D). Previous studies have demonstrated that epithelial-to-mesenchymal transition (EMT) and proapoptotic Bcl-2 family member (BIM) may contribute to primary resistance to EGFR-TKIs [24]. We found that the transfection of miR-200c inhibited the EMT pathway by suppressing ZEB1 and activating E-cadherin and that the expression of BIM was also inhibited (Figure 6D). In addition, the combination treatment with the transfection of miR-200c rendered CL1-5 cells more sensitive to gefitinib (Figure 6E).

### 3.8. The EGFR-Mutation Abundance Is Not Associated with Clinical Outcome

To dismiss the possibility that the EGFR-mutation abundance instead of EV miRNA contributed to the clinical outcome, we estimated the EGFR-mutation abundance by comparing EGFR DNA to internal control (exon 2) using real-time PCR. Eight tissue samples of 16 cases were available for real-time PCR analysis, because the residual tissue of the other six specimen was too small for real-time PCR study. In addition, two MPE samples were also not suitable for real-time PCR study. We found there is no correlation between EGFR-mutation abundance and PFS, and the EGFR-mutation abundance were also not significant between good responders (PFS > 12 M) or poor responders (PFS < 6 M) (Figure 7A, Appendix A). We also performed immunohistochemistry (IHC) staining [25] in two specimens from patients harboring Exon 19Del mutation but with different response to EGFR-TKI. The expression of EGFR-mutant specific protein is not different between a good responder and poor responder (Figure 7B).

## 4. Discussion

In the current study, we first verified the EV from EGFR-mutant cells can be transferred to EGFR wild-type cell and therefore change their sensitivity to EGFR-TKI in vitro and in vivo. When we performed a comparative analysis of EV miRNA content from EGFR-mutant and wild-type cell, we identified the miRNAs which mediate EGFR downstream signaling pathways, such as the PTEN-AKT, Stat3, and Erk pathways. We further compared these miRNAs to the differentially circulating miRNAs from patients with different response to EGFR-TKI. One of these miRNAs, miR-200c, not only suppressed EGFR downstream pathway but also inhibited the pathway associated with EGFR-TKI resistance such as BIM and EMT pathway. We also used real-time PCR to show that EGFR-mutation abundance did not affect PFS of EGFR-mutant patients and implied the contribution of response to EGFR-TKI in heterogeneous EGFR-mutant NSCLC may come from EV miRNA.

Extracellular vesicles such as exosomes are powerful mediators of intercellular communication between cancer cells and tumor microenvironments. In lung cancer, a recent study demonstrated that the treatment of recipient cancer cells with EVs from gefitinib-resistant PC9 cells increased the phosphorylation of AKT and mTOR and enhanced proliferation, invasion, and drug resistance to gefitinib-induced apoptosis [10]. Another study revealed that exosomes derived from gefitinib-treated PC9 cells decreased the antitumor effects of cisplatin [9]. Conversely, exosomes derived from cancer cells may also sensitize cancer cells to chemotherapy, especially after exposure to chemotherapy or irradiation. In a melanoma animal model, exosomes isolated from postirradiated melanoma cells were found to contain key damage-associated molecular patterns (DAMPs), while the intratumoral injection of exosomes from irradiated cells or tumors significantly delayed tumor growth in an in vivo engraftment model [26]. In breast cancer cells, treatment with topotecan (TPT) inhibited tumor growth in tumor-bearing mice, which was accompanied by the infiltration of activated DCs and CD8+ T cells. That study further demonstrated that TPT-treated cancer cells could activate STING signaling and induce antitumor immunity by activating DC responses to exosomal DNA derived from tumor cells [27]. Our study suggested that exosomal cargos can be transferred between cancer cells with different genetic backgrounds and that these cargos possess bioactivity that can change the EGFR-TKI sensitivity of EGFR wild-type cells.

Valadi et al. [6] first proposed the concept of “exosomal shuttle RNA” (esRNA). They demonstrated that transferred exosomal mRNA can be translated after entering another cell because exosomal RNA from mast cells can be transferred to other mouse cells, and new mouse proteins were found in the recipient cells after exosome uptake [6]. Moreover, by adding RNase into the culture media of HT-29 cells and into fecal homogenates, Koga et al. [28] found that exosomal miRNAs were protected from RNase by the exosomes, whereas free miRNAs were degraded by RNase. In ovarian cancer, miR-6126 was found to be consistently overexpressed in exosomes from ovarian cancer cell lines, and higher levels of miR-6126 were associated with longer survival and better prognosis in ovarian cancer patients, supporting the role of EV miRNAs in predicting survival [29] and keeping in mind that miRNAs affect multiple genes within a single cell and also affect the gene expression of adjacent cells by binding to the 3′UTR of their target mRNA. In our study, we identified an exosomal miRNA panel with differential expression patterns between EGFR-mutant cells and wild-type cells (Figure 5A, Appendix A). Most of the miRNAs belonged to the miR-200 family, and subsequent pathway analysis showed that the involved pathways were the main downstream pathways of EGFR (Figure 5B,C) [30]. When we compared these miRNAs with those identified in circulating EVs from patients with a good responses or poor responses to EGFR-TKIs (Figure 6A, Appendix A), we found that miR-200a and miR-200c were upregulated in the responders. Conversely, miR-210 and miR-758 were downregulated in the responders. MiR-210 and miR-758 have been found to be important prognostic factors of lung cancer; recently, an overall pooled meta-analysis indicated that there were higher levels of miR-210 expression in NSCLC cancerous tissue than in normal control tissue and that the overexpression of miR-210 was associated with poor outcome [31]. In addition, the overexpression of miR-758 inhibited the proliferation, migration, invasion, and cell-cycle progression of NSCLC cells while also stimulating their apoptosis by negatively regulating high-mobility group box (HMGB) [32]. However, the relationship between exosomal miR-210, miR-758, and gefitinib sensitivity in NSCLC has not been previously reported. A previous study on miR-200a and miR-200c showed that miR-200a in NSCLC cells significantly downregulated both EGFR and c-Met levels and severely inhibited cell migration and invasion. Moreover, in gefitinib-resistant cell lines, miR-200a expression was able to render the cells much more sensitive to drug treatment [33]. Recent studies have emphasized the role of miR-200c in predicting lung cancer prognosis, but the results of those studies have been contradictory since miR-200c has been reported to have oncogenic and tumor-suppressive functions [34,35,36]. Another study reported that miR-200c overexpression is associated with better responses to EGFR-TKIs in patients with EGFR wild-type NSCLC [35]. However, the role of exosomal miR-200c in mediating gefitinib sensitivity in heterogeneous NSCLC remains unknown. Our study was the first to identify the role of EV miRNAs in mediating EGFR-TKI sensitivity in heterogeneous EGFR-mutant NSCLC. We found that miR-200c not only inhibited the downstream signals of the EGFR pathways but also affected the pathways involved in primary resistance to EGFR-TKIs, such as epithelial-to-mesenchymal transition (EMT) and BCL2-like 11 (BIM)-mediated apoptosis [37]. However, there were some limitations to our study. First, although we proved that the EVs from PC9 cells enhanced gefitinib sensitivity in some EGFR wild-type cells, we are not sure if this finding applies to EGFR L858R mutant cells. Our study also did not provide the evidence that PC9-derived EV enhanced the sensitivity of EGFR wild-type to Osimertinib, which has been approved as the first line treatment in EGFR-mutant NSCLC by the FDA and EMA [38]. However, in our validation cohort, 7 of the 10 patients harbored L858 mutations; therefore, the significance of the identified EV miRNAs is promising in this regard. Second, our study focused only on the role of miRNAs from EVs and on the mechanism of miR-200c. Proteomic analyses of EVs and mechanistic studies of other potential miRNAs were not performed. Third, we used a higher concentration of gefitinib for combination treatment with EV (0.1 μM) or miR-200c (5 μM) compared to IC50 in PC9 in vitro [39]. However, when we compared the viability of CL1-5 cells treated with a combination of 5 μM gefitinib and miRNA (50.8%, Figure 6E) to that of cells treated with 10 μM gefitinib alone (85.3%, Figure 2A), cell viability significantly decreased in the combination treatment group. Finally, the miRNAs we identified were different from those identified by other studies to predict responses to EGFR-TKIs [33,34]. However, one recent study demonstrated that EV-incorporated and whole-plasma cell-free miRNA profiles were clearly different in prostate cancer [40]. Moreover, unlike EV miRNAs, plasma cell-free miRNAs were easily degraded under different temperatures. Therefore, EV miRNAs serve as potential biomarkers based on their stability under various conditions [41]. That said, whether EVs are a better source for testing these miRNAs as lung cancer biomarkers than whole plasma requires validation in a larger cohort.

## 5. Conclusions

Our study revealed a novel mechanism explaining why patients with heterogeneous EGFR-mutant NSCLC can still respond to EGFR-TKIs. The exposure of EGFR wild-type cells to EGFR-mutant cell EVs changed the sensitivity to EGFR-TKI treatment, possibly via the transmission of EV miRNAs that suppressed the Stat3, Erk, EMT, and BIM pathways and then induced cell apoptosis. Furthermore, we found that these miRNAs can be detected in blood and are significantly differentially expressed between good responders and poor responders. Based on these results, we envision that exosomal microRNAs may not only serve as predictors of the response to EGFR-TKIs but also provide an alternative approach for lung cancer interventions in heterogeneous EGFR-mutant NSCLC.

## Figures and Tables

**Figure 1 biomedicines-09-00243-f001:**
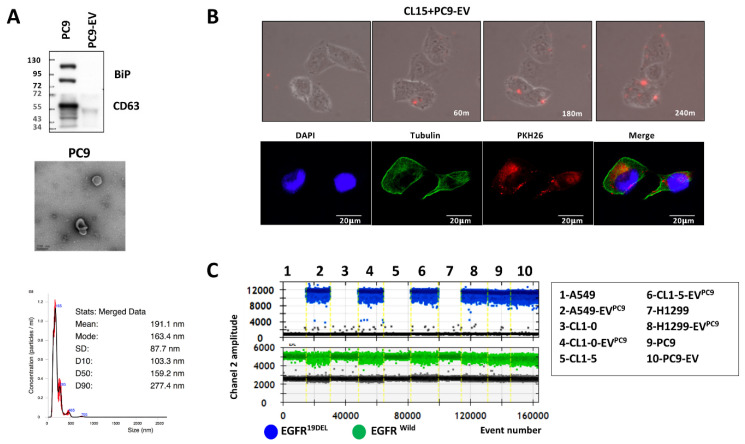
The characteristics of extracellular vesicles and their transfer from PC9 cells to CL1-5 cells. (**A**) The presence of CD63 in isolated extracellular vesicles from PC9 cells was analyzed by Western blot (upper). Extracellular vesicles were observed by transmission electron microscopy (middle). Scale bar, 200 nm. NanoSight particle analysis of extracellular vesicles derived from PC9 cells (lower). (**B**) Time-lapse images following the addition of PKH26-labeled extracellular vesicles to CL1-5 cells and immunostaining showing red fluorescence (EV-PKH26) inside tubulin-positive cells. (**C**) epidermal growth factor receptor (EGFR)-mutant DNA was detected using droplet digital PCR (ddPCRTM; Bio-Rad) in PC9 cells, extracellular vesicles (EVs) from PC9 cells, and EGFR wild-type cells after treatment with PC9 EVs.

**Figure 2 biomedicines-09-00243-f002:**
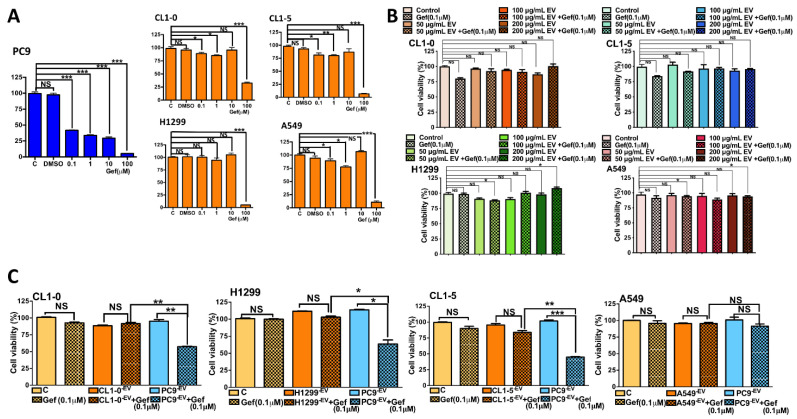
EVs from PC9 cells increased the sensitivity of EGFR wild-type cells to gefitinib in vitro. (**A**) MTT assay of PC9, CL1-0, CL1-5, H1299, and A549 cells in response to gefitinib. The cells were treated with gefitinib for 72 h. (**B**) CL1-0, CL1-5, H1299, and A549 cells were treated with gefitinib and titrated control EVs (200 μg/mL) for 72 h. In addition, (**C**) CL1-0, CL1-5, H1299, and A549 cells were treated with gefitinib and EVs (200 μg/mL) from PC9 cells or control EVs for 72 h (* *p* < 0.05, ** *p* < 0.01, and *** *p* < 0.005).

**Figure 3 biomedicines-09-00243-f003:**
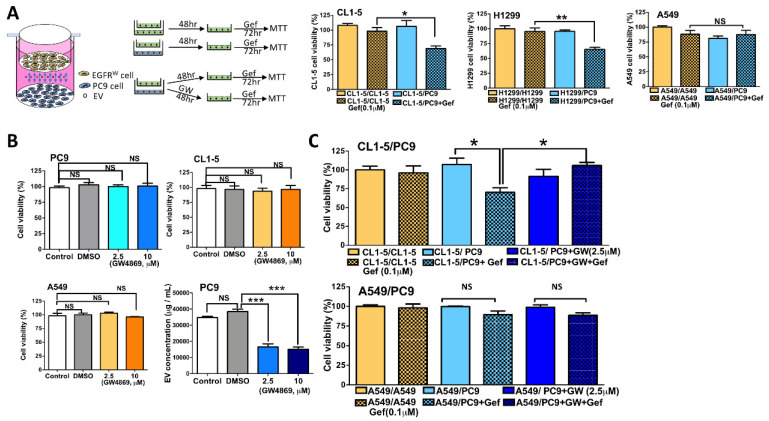
The sensitivity of EGFR wild-type cells increased in coculture with PC9 cells but decreased if PC9 cells were cocultured for 48 hours in a coculture system. (**A**) A schematic diagram illustrating the design of the coculture experiments. (**B**) EGFR wild-type cells were treated with gefitinib for 72 h after coculture with PC9 cells, and cell viability was examined by MTT assay. (**C**) The effect of GW4869 on cell viability was examined by MTT assay, and the effect of EVs was determined by ELISA by measuring total protein. After adding GW4869 to the coculture system, EGFR wild-type cells were treated with gefitinib for 72 h, and cell viability was examined by MTT assay (* *p* < 0.05, ** *p* < 0.01, *** *p* < 0.005).

**Figure 4 biomedicines-09-00243-f004:**
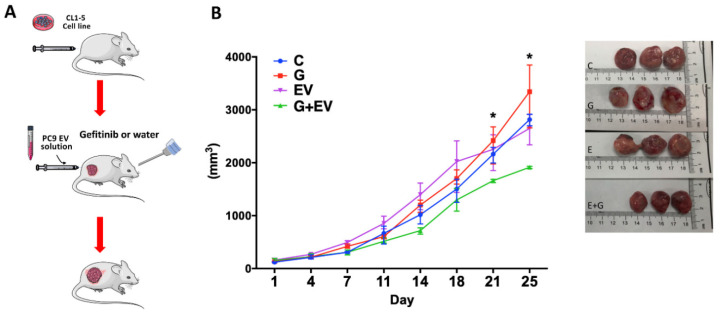
EVs derived from EGFR-mutant cells delayed EGFR wild-type tumor growth in vivo. (**A**) A schematic diagram illustrating the design of the animal model. (**B**) Dynamic tumor growth and treatment responses to (C) control, (G) gefitinib (50 mg/kg/day), (EVs) extracellular vesicles (20 μg, intratumor injection twice per week), and (G+EVs) gefitinib + extracellular vesicles (* *p* < 0.05).

**Figure 5 biomedicines-09-00243-f005:**
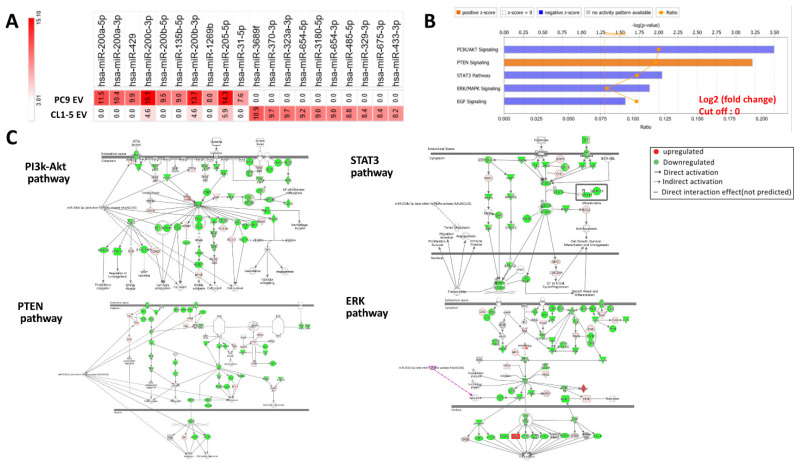
Different micro-RNA expression profiles between EVs from EGFR-mutant cells and EVs from EGFR wild-type cells and associated regulatory signal pathway. (**A**) Differential miRNA expression in EVs from PC9 cells compared to those from CL1-5 cells. (**B**) A miRNA regulatory signal network analysis was performed using ingenuity pathway analysis (IPA) software. (**C**) This gene network displayed the top ranked network found associated with the mi200. Red and green tags represent the upregulated and downregulated genes, respectively.

**Figure 6 biomedicines-09-00243-f006:**
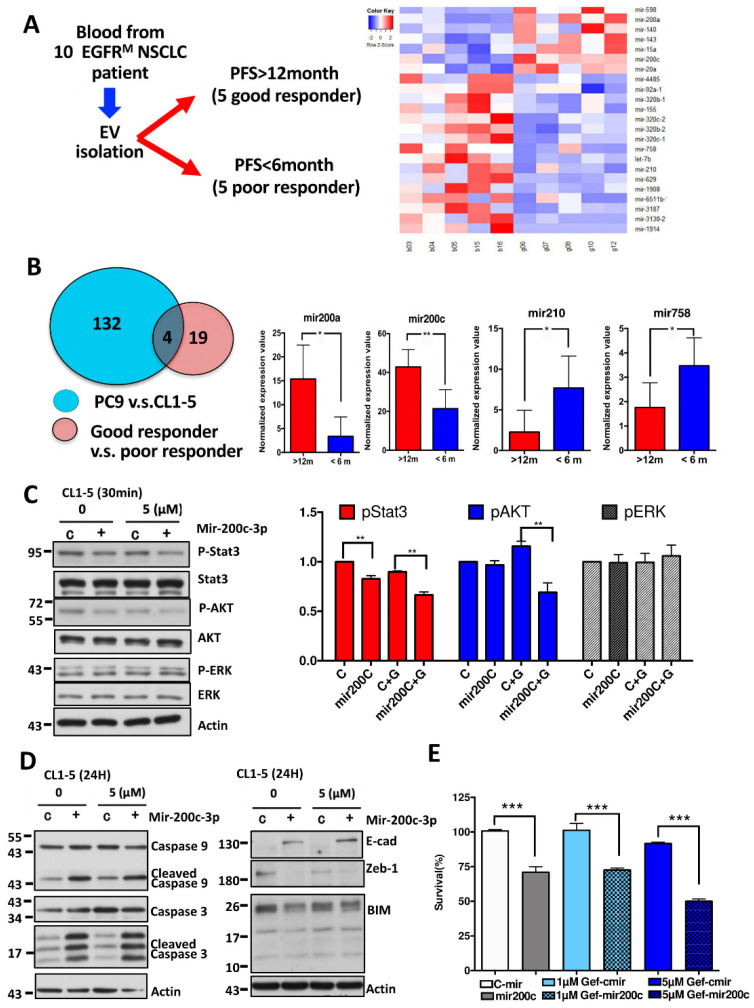
Differential miRNA expression from circulation EVs from patients with different responses to EGFR-TKI treatment corresponded to that from cell lines EVs. (**A**) Flowchart showing the collection of blood exosomal miRNAs from patients with different responses to EGFR-TKIs. Identification of differentially expressed miRNAs in responders and nonresponders to EGFR-TKIs. (**B**) Identification of miRNAs that are both differentially expressed between responders and nonresponders to EGFR-TKIs and between EVs from PC9 cells and CL1-5 cells. (**C**) CL1-5 cells were transfected with miR-200c and treated with gefitinib, and the pStat3, Stat3, pAkt, Akt, pErk, and Erk protein levels were detected with a Western blot kit. (**D**) The changes in apoptosis-related proteins, including caspase-3 and caspase-9, after miR-200c transfection with gefitinib treatment were determined using Western blotting. The combination of miR-200c and treatment with gefitinib affected EMT-related proteins (E-cadherin and Zeb1) and BIM protein. (**E**) Histograms showing the enhanced efficacy of gefitinib in CL1-5 cells transiently transfected with miR-200c as determined by MTT assays (* *p* < 0.05, ** *p* < 0.01, *** *p* < 0.005).

**Figure 7 biomedicines-09-00243-f007:**
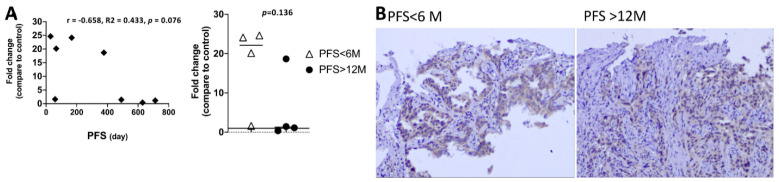
The EGFR-mutation abundance analysis using real-time PCR and immunohistochemistry staining. (**A**) To calculate the relative expression of the EGFR-mutant mRNA normalized to internal control (exon 2), the average of target Ct was subtracted from the average of Exon 2 Ct (ΔCt). The amount of EGFR-mutant mRNA normalized to an endogenous reference and relative to a calibrator (fold-change) is given by 2^−ΔΔCt^ where the calculation of ΔΔCt involves subtraction by the ΔCt calibrator value. Assuming that ΔCt = 3.32 corresponds to a 10-fold difference of expression between exon 2 and EGFR. There was no correlation between relative EGFR-mutation abundance and PFS of patients receiving EGFR-TKI analyzed by Pearson’s correlation coefficient (r) method. In addition, EGFR-mutation abundance in a poor responder (PFS < 6 months) and good responder (PFS > 12 months) are also not different. Student’s t-test was used. (**B**) Immunohistochemical staining of EGFR 19Del specific protein showed similar intensity in tumors from patients with different response to EGFR-TKI (Magnification, 200×).

**Table 1 biomedicines-09-00243-t001:** Clinical characteristics of enrolled patients.

Patient/Age Sex/Smoking	ECOG ^1^	PFS ^2^	Mutation	Stage	PD Location	Best Response	EKI
1/59/M/−	0	2.2	L858R	T3N3M1b	MPE ^3^	PD ^4^	Erlotinib
2/75/F/−	1	5.2	19Del	T4N3M1a	lung	SD	Erlotinib
3/82/M/−	0	4.9	L858R	T3N2 M1a	lung	SD	Afatinib
4/56/M/+	0	2.0	L858R	T4N3M1b	intestine	PD	Erlotinib
5/53/M/+	1	1.0	L858R	T4N3M1b	lung	PD	Afatinib
6/53/F/−	1	28.8	L858R	T3N2 M1a	lung	SD	Gefitinib
7/58/M/+	0	19.8	L858R	T1N3M1b	brain	PR	Erlotinib
8/80/M/+	1	16.0	19Del	T4N3M1b	liver	PR	Gefitinib
9/64/M/+	1	12.6	19Del	T4N3M1b	brain	PR	Erlotinib
10/73/F/−	1	17.5	L858R	T4N3M1b	MPE	PR	Erlotinib

^1^ ECOG; Eastern Cooperative Oncology Group performance status, ^2^ PFS; Progress-free survival (months), ^3^ MPE; Malignant pleural effusion, ^4^ PR = partial response; SD = stable disease; PD = progressive disease.

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
