# Peer review of "Extracellular Vesicle miR-200c Enhances Gefitinib Sensitivity in Heterogeneous EGFR-Mutant NSCLC"

_biomedicines, 2021, doi:10.3390/biomedicines9030243_

Round 1
Reviewer 1 Report
The paper presents the role of extracellular vesicles, containing a series of miRNA, in increasing the sensitivity to gefitinib in NSCLC. The findings furtherly prove the important role EV can play in influencing cancer progression and therapy. A side by side comparison between in vitro assays and patient derived studies is performed. Data are sufficiently well presented but a series of details should be added in order strengthen the solidity of the story. Please find attached a series of suggestions:
- In reviewer’s opinion the abstract need to be a bit improved by including few introducing information, this is supposed to help the reader in orienting himself in the specific context. Also, some acronyms need to be expanded at least at their first appearance in the text (this applies to the rest of the manuscript as well).
- Please in line 63 convert protein in proteins.
- Method section need to be integrated with a series of details, this also in order to allow others to follow the specific procedures and possibly reproduce the proposed experiments. In the upcoming comments the reviewer will highlight the main portions needing this integration.
- Line 204, please remove the article “the” from the title.
- Paragraph 3.1: As for reviewer understanding EV are firstly isolated from PC9 than characterized and than stained and used to treat CL1-5. Please reorganize this paragraph by respecting this order.
- Please magnify scale bars in figure 1b.
- Please explain better the data presented in figure 1c. Also, add a description of the reported plot. Please explain how the treatment with EV can influence the EGFR-mutant DNA copy number and add details about the technique in method section.
- Paragraph 3.2: Please include IC50 calculation.
- Please indicate the concentration of gef used for the data in figure 2b and create a paragraph in method section about cell viability assays.
- It is not clear which EGFR wild-type cells EV were used for generating data in figure 2b.
- The whole section 3.2 need to be presented more clearly and details need to be included, please also include in method section as well.
- Figure 3: just out of curiosity, reviewer wonders if there is any reason why CL1-1 is not present in this data set. Please include in methods a paragraph dedicated to the co-cultures.
- Figure 4: was any survival curve generated? Please include details on EV administration also in results (2/week). Very important: Methods and Results section are not consistent in the amount of EC used. In method it is indicated 20ug in results 2ug, please amend one of the two sections with the correct mass.
Reviewer 2 Report
Chien-Chung Lin et al. undertook an excellent work in order to find the role of extracellular vesicles (EVs) in anti-apoptotic signals between drug-resistant and drug-sensitive cells. They performed co-culture experiments, in vivo experiments, they recollect human samples, etc. They demostrate why heterogenous EGFR-mutant NSCLC patients still respond to EGFR-TKIs. So, in my opinion, the presented manuscript is completed and very hard-worked.
I really enjoyed reading their manuscript and I find no major issues to be concerned. As to the minor ones – they are rather of editorial type, manageable in the editorial correction process.
Author Response
We would like to express our sincere appreciation for your full support for our study and contribution in reviewing the manuscript.